# Metallothionein Expression as a Physiological Response against Metal Toxicity in the Striped Rockcod *Trematomus hansoni*

**DOI:** 10.3390/ijms232112799

**Published:** 2022-10-24

**Authors:** Rigers Bakiu, Sara Pacchini, Elisabetta Piva, Sophia Schumann, Anna Maria Tolomeo, Diana Ferro, Paola Irato, Gianfranco Santovito

**Affiliations:** 1Department of Aquaculture and Fisheries, Agricultural University of Tirana, 1000 Tirana, Albania; 2Department of Biology, University of Padua, Via U. Bassi 58/B, 35131 Padua, Italy; 3Department of Cardiac, Thoracic and Vascular Science and Public Health, University of Padua, 35128 Padua, Italy; 4Department of Pediatrics, University of Missouri-Kansas City, Kansas City, MO 64108, USA

**Keywords:** Antarctica, antioxidants, cadmium, copper, fish, metallothioneins

## Abstract

Metal bioaccumulation and metallothionein (MT) expression were investigated in the gills and liver of the red-blooded Antarctic teleost *Trematomus hansoni* to evaluate the possibility for this species to face, with adequate physiological responses, an increase of copper and cadmium concentrations in its tissues. Specimens of this Antarctic fish were collected from Terra Nova Bay (Ross Sea) and used for a metal exposure experiment in controlled laboratory conditions. The two treatments led to a significant accumulation of both metals and increased gene transcription only for the MT-1. The biosynthesis of MTs was verified especially in specimens exposed to Cd, but most of these proteins were soon oxidized, probably because they were involved in cell protection against oxidative stress risk by scavenging reactive oxygen species. The obtained data highlighted the phenotypic plasticity of *T. hansoni*, a species that evolved in an environment characterized by naturally high concentrations of Cu and Cd, and maybe the possibility for the Antarctic fish to face the challenges of a world that is becoming more toxic every day.

## 1. Introduction

The Antarctic environment has unique characteristics related to its distance and isolation from other continents of our planet. Anthropogenic contamination is considered negligible even if contaminants can reach Antarctica by long-range atmospheric transport [1].

Antarctic marine organisms evolved in this environment, isolated for 10–12 million years and exposed to peculiar physical and chemical conditions, such as a very low and constant temperature and high oxygen concentration [2]. Such conditions probably affected the adaptive metabolic strategies during the evolution of these organisms [3] and, in particular, the physiological defence systems against the risk of oxidative stress [4,5].

One of the main characteristics of Antarctic seawater is a natural occurrence of high cadmium (Cd) concentrations, about 70 ng L^−1^ in the soluble fraction and 0.05–0.49 μg/g dry mass in surface sediments along the coast of Terra Nova Bay, Ross Sea [6,7].

Antarctic seawaters are also characterized by a natural occurrence of high copper (Cu) concentrations, about 150 ng L^−1^ [6].

Due to these elevated environmental metal levels, organisms may accumulate metals, which penetrate their tissues by various mechanisms, according to the chemical speciation of the metal. The main pathway for metal uptake in fish seems to be through the gills and intestines, but the relative extent of these routes varies, partly depending on the chemical and physical characteristics of water and sediments [8]. Antarctic vertebrates can accumulate metals also feeding on molluscs and epibenthic crustaceans because these prey frequently have high tissue metal concentrations [7,9].

It is well-known that metal ions widely interact with biological molecules. Cd has a high affinity for the sulphydryl cysteine groups and competes against zinc (Zn) and Cu for the structural and active sites of various enzymes, thus impairing their catalytic activities [10]. Cd also exerts several toxic effects at both cellular and systemic levels [11]. On the other hand, essential trace metals, such as Cu, are required in various physiological and metabolic processes, but they become toxic at excessive concentrations, damaging plasma membrane and other cell components [12].

Metal ions can cause severe problems in fish by producing reactive oxygen species (ROS). These substances are a normal by-product of oxidative metabolism, and generally, a higher rate of ROS generation has been reported in fish and aquatic invertebrates [13]. On the other hand, mitochondria of true endotherms, such as mammals, seem to produce lower amounts of ROS than fish [14]. However, the rate of ROS formation can vary from species to species and can also greatly increase in response to various physiological and pathological conditions, causing oxidative stress and damaging DNA, proteins, and lipids [15,16]. Regarding metals, Cu is one of the active redox components that promote the redox response to ROS. Cu can specifically operate as a catalyst in the Fenton reaction, promoting the transformation of superoxide anion and hydrogen peroxide into hydroxyl radical, the most damaging ROS and the primary cause of oxidative stress. [17]. On the other hand, redox-inactive metals such as Cd show their toxic effects via bonding to sulphydryl groups of proteins and depletion of glutathione [18].

Organisms developed metallothioneins (MTs), a family of widely distributed, low-molecular-weight metal-binding proteins (6–18 kDa), as the first line of defence against metal toxicity to reduce the issues associated with metal exposure [19,20]. These proteins are characterized by an unusually high cysteine content (30%) and a lack of aromatic amino acids and histidine [21]. Their biosynthesis may be induced in tissue by various stimuli, especially metal ions or several stress conditions, having many functions, such as the regulation of essential metal content, the detoxification of essential and non-essential metals, and the non-enzymatic scavenging of ROS [22]. Tissue expression of MTs in fish primarily occurs in the liver, kidney, and gills [23].

One of the main scientific questions about the physiological adaptation of Antarctic organisms is whether they have evolved acclimatization capacities towards a variation of the environmental metal concentrations given that they evolved under a significant selective pressure represented by the elevated concentrations of these chemical elements. In particular, in this paper, we aimed to verify whether and how an increase in the tissue concentrations of Cu and Cd can be reflected in the implementation of the gene expression of MTs. *Trematomus hansoni*, an Antarctic teleost widespread in the coastal marine areas of the ice continent, was chosen as the experimental organism and used in exposure experiments under controlled laboratory conditions. The data of metal accumulation in gills and the liver have been correlated with the MT gene expression and evaluated at both transcriptional and post-transcriptional levels.

Knowledge of *T. hansoni* physiology is very scarce and almost exclusively limited to the anti-freezing properties of its body fluids, a feature shared with other species of Antarctic fish [24]. This species showed only minor differences in antioxidant defences compared to other notothenioids, with strong antioxidant capacity but restricted catalase activity [25], suggesting that additional proteins are crucial for the defence against ROS.

## 2. Results and Discussion

The exposure to 1.57 µM Cu or 0.89 µM Cd was non-lethal for the experimental specimens, confirming the results of the experimentations performed on fish of the same genus [26]. Cu treatment led to a statistically significant accumulation of this metal (*p* < 0.001) in both gills and the liver, with an increase of 53% and 61%, respectively (Figure 1a). The accumulation of Cd was much more consistent in percentage (Figure 1b). In fact, after treatment with this metal, the concentrations measured in the liver almost quadrupled, while in the gills, they became about 100 times higher than controls (*p* < 0.001), in which the Cd concentration was close to zero.

These results can be correlated to the different degrees of metal absorption in different tissues [27]. It is also possible that the higher percentage accumulation of Cd is related to the fact that it is a non-essential metal. In fact, for non-essential metals, cells favour a system of detoxification based on chelation (and therefore storage in molecular structures that reduce its solubility and bioavailability) rather than on excretion, having no membrane transport systems that are certainly present for Cu, which is instead an essential metal [28,29].

Another possibility is that the greater percentage increase in the gills results from acute exposure to high concentrations of this metal. This effect should be emphasized in gills because they are directly exposed to the external environment and lined with a simple epithelium, which is linked to diffusional processes toward body fluids. The results confirm that the liver is an essential organ in detoxifying xenobiotics, accumulating them in greater quantities than other organs and tissues, even in Antarctic fish. For example, Dalla Riva et al. [30] determined higher Cd concentrations in the liver than in the white muscles and spleen of *Trematomus bernacchii*. Santovito et al. [31] found a more significant accumulation of Cu and Cd in the liver compared to other tissues (gills, heart, white muscle) in both *T. bernacchii* and *Trematomus newnesi*. Furthermore, *T. bernacchii* experimentally treated with different metals showed a significant increase in the hepatic concentration of these elements [32].

As a consequence of metal accumulation, there is an increase in the transcription of genes encoding MTs but only in the liver and exclusively for the MT-1 isoform (Figure 2a), with mRNA concentrations showing a relatively small increase of 15% after exposure to Cu and 10% after treatment with Cd (*p* < 0.05).

This result is partially confirmed by the literature data, as Cd exposure studies using the icefish *Chionodraco hamatus* showed that MT-1 transcript was preferentially accumulated in the liver [33]. However, it is generally assumed that isoform 1 plays a significant role in the detoxification of Cd [27].

Recent research in *Trematomus eulepidotus* suggests that the preference for the production of the various MT isoforms may not be tissue-specific and instead may depend on the inducing metal and the species under consideration [34]. More likely, structural and functional differences in the promoter regions of the genes encoding these proteins are responsible for the differential expression of the two MT isoforms [35].

A rather singular result was obtained by measuring the tissue levels of MT with the silver saturation method [36]. As can be seen in Figure 3a, the specimens exposed to both Cu and Cd do not show an increased concentration of MT either in the gills or in the liver, and even the levels are statistically lower in the treated specimens compared to the controls (*p* < 0.05). Given the role played by MTs as metal-binding proteins, it was unlikely that an increase in the accumulation of Cu and Cd would lead to a physiological response characterized by a reduction in the presence of these proteins at the cellular level. The existence of these proteins could not be emphasized using the silver saturation method because they were partially oxidized, which prevented us from confirming the theory that the biosynthesis of MTs had taken place. In fact, by applying Santovito et al.’s method [37], which can also measure oxidized MTs, an increase in MT expression is evident (*p* < 0.05), in particular in response to Cd (Figure 3b). This result is in line with what was previously highlighted by the analysis of metal accumulation. The only exception is represented by the gills of the specimens exposed to Cu, in which the TM levels are comparable to those measured in the controls. In this particular circumstance, other chelating molecules, such as glutathione (GSH), may play a more critical role.

It is well-known that this tripeptide plays an important protective role against the toxic effects of metals, acting as a primary chelating molecule in an early phase of the intracellular accumulation of these elements, as indeed occurs during acute exposure [38,39]. Furthermore, it may not be surprising that this occurs precisely in the gills of specimens exposed to Cu. Cu is a metal with redox properties and may be involved in Haber-Weiss and Fenton reactions, producing ROS [40,41]. GSH is known to form stable GSH-Cu (I) complexes, preventing further redox cycling and the generation of free radicals, and this may explain the complete protection afforded by GSH against the effects of Cu [35]. In addition, this preventative function in creating an oxidative stress situation is crucial in an organ such as the gill, where high partial pressures of oxygen are naturally present and favour the generation of ROS.

Among the protein alterations affecting MTs, whose characteristics are known, is the oxidation of these molecules with the formation of disulphide bridges between the thiol groups of the cysteines [42]. This protein modification is linked to the antioxidant defence function characteristic of MTs [43,44]. Under our experimental conditions, this is probably determined by an increase in the rate of ROS formation, the production of which is notoriously enhanced by the presence of an excess of metals in the cell [17].

Another peculiar result is that the concentrations of MTs in the liver are not correlated with their mRNA levels. Since the rate of protein biosynthesis is higher than the rate of messenger transcription, it is possible that in the controls is a relatively high mRNA concentration, which is not fully translated to proteins yet. This result is widely documented in the literature, and many authors attribute it to a post-transcriptional control on MT synthesis, first identified in the rat liver [45]. This regulation appears to depend on the development of stress granules (SGs), which are cytoplasmic foci, where the messengers can be stored and translated later. [46]. We have frequently emphasized this characteristic in species that are exposed to stressogenic environments but that are not experiencing acute stress [47,48], such as Antarctic fish [49,50,51], which enables the tissues to react to acute stress very quickly when it suddenly arises. The occurrence of stress granulates in the liver of *T. hansoni* experimentally exposed to metals is a possible explanation for our results. 

Nucleation proteins are involved in SG formation, such as the T-cell-restricted intracellular antigen (TIA) proteins, TIA-1, and TIA-1-related protein (TIAR), which both self-associate to promote the growth of SGs, directly binding target RNAs [52]. Recently, we characterized these proteins and their expression concerning the expression of anti-stress proteins in *C. hamatus* and *T. bernacchii*. Preliminary data indicated that, in both species, high levels of expression of the messenger of TIA-1 correspond to low levels of biosynthesis of antioxidant enzymes, such as peroxiredoxins, supporting the hypothesis of a post-transcriptional control operated by stress granules [53]. We obtained similar results studying SG proteins in the solitary ascidians *Ciona robusta* experimentally exposed to metals [54].

## 3. Materials and Methods

### 3.1. Ethical Procedures

The sample collection and animal research conducted in this study comply with the Italian Ministry of Education, University, and Research regulations concerning activities and environmental protection in Antarctica and with the Protocol on Environmental Protection to the Antarctic Treaty, Annex II, Art. 3. All experiments were performed under the U.K. Animals (Scientific Procedures) Act, 1986 and associated guidelines; EU Directive 2010/63/EU; and Italian DL 2014/26 for animal experiments.

### 3.2. Experimental Animals

Adult specimens of *T. hansoni* (21.2–24.6 cm, 130–159 g) were collected in the proximity of Mario Zucchelli Station in Terra Nova Bay, Antarctica (74°42′ S, 167°7′ E), and kept in glass aquaria of 180 L (100 × 40 × 45 cm^3^) supplied with aerated seawater at approximately 0 °C (pH 8.1, 10.3 mg O_2_ L^−1^).

After a distressing period of seven days, 10 specimens were randomly distributed in two tanks (five for each tank), where they were exposed to Cu (1.57 µM) or Cd (0.89 µM), sub-lethal doses previously used in similar experimentations on fish of the same genus [29]. This choice was made to be able to make a comparison between different species exposed to the same concentrations of Cu and Cd. Furthermore, the goal was to produce a tissue accumulation of metal sufficient to produce a cellular response. Since the exposure time had to be necessarily reduced due to the limited possibilities granted by the particular laboratory conditions, it was decided to use non-”environmental” concentrations.

Five untreated specimens were used as a control group. After 5 days, all the fish were euthanized (tricaine methanesulphonate, MS-222; 0.2 g L^−1^), and samples of gills and liver tissues were excised, quickly frozen in liquid nitrogen, and stored at −80 °C.

### 3.3. Primers Design, Total RNA Extraction, mt-1, and mt-2 cDNA Synthesis

Primers were designed in the coding regions of the *mt-1* and *mt-2* cDNA sequences previously characterized in *T. hanasoni* and published in the *NCBI* database (GenBank accession numbers FJ870679.1 and FJ870680.1, respectively). Primer sequences were analysed with the IDT Oligo Analyzer tool (https://eu.idtdna.com/pages/tools/oligoanalyzer (accessed on 6 September 2021)). Primer sets are shown in Appendix A. 

Total RNA was purified from tissues using PRImeZOL™ reagent (Canvax, Córdoba, Spain) according to the manufacturer’s protocol. Further purification was performed with 8M LiCl [55] to remove glucidic contaminants. The RNA quantification was performed using the NanoDrop ND-1000 spectrophotometer (ThermoFisher Scientific, Waltham, MA, USA). RNA integrity was assessed by running an aliquot of RNA (1000 ng μL^−1^) on a denaturing gel stain [56]. The cDNA synthesis was performed using a Biotechrabbit™ cDNA Synthesis Kit (Berlin, Germany) at 50 °C for 1 h + 99 °C for 5 min, from 1 μg of total RNA in a 20 μL reaction mixture, containing 2 μL of dNTP Mix (10 mM each), 0.5 μL of RNase Inhibitor, 40 U μL^−1^, 0.5 μL of Oligo (dT) 12–18 (10 μL), 4 μL of 5x Reverse Transcriptase Buffer, 1 μL of RNA Template, 1 μL of RevertUPTM II Reverse Transcriptase, and PCR-grade water up to 20 μL. PCR reactions were performed with 50 ng of cDNA and GRS Taq DNA polymerase (Grisp, Porto, Portugal). The PCR program was the following: 95 °C for 5 min and 40 × (95° for 30 s, Ta for 30 s, 72 °C for 30 s) and final elongation at 72 °C for 5 min.

### 3.4. qRT-PCR Analysis

Real-time qRT-PCR analysis was performed to evaluate the expression of *mt-1* and *mt-2* mRNAs. cDNAs for both target genes were amplified with the specific primers reported in Appendix A. The housekeeping gene *gapdh* was amplified with species-specific primers (Appendix A) to control for variation in the efficiency of cDNA synthesis and PCR amplification reactions. qRT-PCR amplifications were carried out using the qPCRBIO SyGreen Mix Separate-ROX kit (PCR Biosystems, Wayne, PA, USA) and the following program: 95 °C for 2 min, 40 × (95 °C for 20 s and 60 °C for 60 s), and then the dissociation stage 95 °C for 15 s, 60 °C for 1 min, 95° for 15 s, and 60 °C for 15 s.

### 3.5. Estimation of Metal and Metallothionein Concentrations

Polytron homogenized portions of the tissues in 4 vol g^−1^ of tissue of 0.5 M sucrose, 20 mM Tris–HCl buffer pH 8.6, supplemented with 0.006 mM leupeptin, 0.5 mM PMSF (phenylmethylsulphonyl fluoride) as an antiproteolytic agent, and 0.01% β-mercaptoethanol as a reducing agent. The homogenates were centrifuged at 48,000× *g* for 60 min at 4 °C to obtain the cell-free extracts.

Cu and Cd contents were determined in cell-free extracts by atomic absorption spectroscopy using a PerkinElmer (Waltham, MA, USA) 5100 graphite furnace atomic absorption spectrometer. The instrumental conditions applied were according to the PerkinElmer manual. Hollow cathode lamps for each analysed metal were used as radiation sources. For Cu and Cd, the working wavelengths were 324.754 nm and 228.802 nm, respectively. The instrument was calibrated manually by aspirating the prepared working standards of the cations of interest (1000 mg L^−1^ stock solutions of metals in deionized water) one by one into the flame. The samples were then also aspirated manually into the flame for atomization. Control blank solution on reagents and equipment revealed insignificant contamination of samples. Values were expressed as ng of single metal/mg of total proteins assayed by the Folin phenol reagent method [57] using bovine serum albumin as standard.

MT concentration was determined in the cell-free extracts by the silver saturation method [36]. Briefly, 0.5 mL of supernatant was mixed with 1.0 mL of 0.5 M glycine buffer, pH 8.5, and 1.0 mL of 20 ppm AgNO3 in 0.5 M glycine buffer, pH 8.5. After standing at room temperature for 15 min, to lead Ag^+^ to saturate the metal binding sites of MT, 100 μL of bovine haemolyzed erythrocytes were added. Then, the mixture was boiled for 2 min to cause denaturation and precipitation of proteins (but non-thermostable MTs) and centrifuged at 4000× *g* for 5 min to remove Ag^+^ bound to haemoglobin and other proteins from the solution. The hemolysate addition, boiling, and centrifugation steps were repeated twice. The MT concentrations were calculated based on the amount of Ag^+^ that remained in the solution because of bonds to MTs. Ag^+^ content was determined by atomic absorption spectroscopy as previously described. The working wavelength was 328.068. To discriminate between reduced and oxidized MTs, Santovito et al.’s method [37] was applied. Briefly, 3.0 mL of cell-free extract were mixed with 2-mercaptoethanol (a reducing agent) at a final concentration of 10 mM and incubated at room temperature for 15 min and then 2 h after adding ZnCl_2_ at a final concentration of 6 mM. Both these incubations were performed in anaerobic conditions, gurgling nitrogen through the solution. The MT concentrations were determined as previously described. As previously described, the amount of MTs was normalized against total soluble cell proteins.

### 3.6. Statistical Analyses

Statistical analyses were performed with the PRIMER statistical program (PRIMER-e, Auckland, New Zealand). One-way ANOVA was followed by the Student–Newman–Keuls test to assess significant differences (*p* < 0.05). The data were expressed as the average of five analysed specimens ± standard deviation (SD).

## 4. Conclusions

The obtained data highlighted the phenotypic plasticity of *T. hansoni*, a species that evolved in an environment characterized by naturally high concentrations of Cu and Cd, and suggested the possibility for the Antarctic fish to face the challenges of a world that is becoming more toxic every day. In this physiological characteristic, metallothioneins play a fundamental role in performing the function of both metal-chelating molecules and non-enzymatic antioxidants. Indeed, this latter function is integrated with other proteins that play a role in protection against oxidative stress, such as antioxidant enzymes, whose expression in Antarctic fish also has peculiar adaptive characteristics.

Furthermore, our results also represent an essential contribution to the achievement of the objectives for establishing the marine protected area (MPA) of the Ross Sea, providing functional data useful for biological assessments for the preliminary and initial monitoring of the ecosystem of the new MPA.

It will be necessary to implement the knowledge of the physiological responses against environmental stress that other species belonging to all kingdoms can carry out against the toxicity of xenobiotic substances. This will certainly be the main goal of future research activities: to have a more detailed big picture not only within the group of Antarctic fish but also of the entire food web of the Antarctic ecosystem.

## Figures and Tables

**Figure 1 ijms-23-12799-f001:**
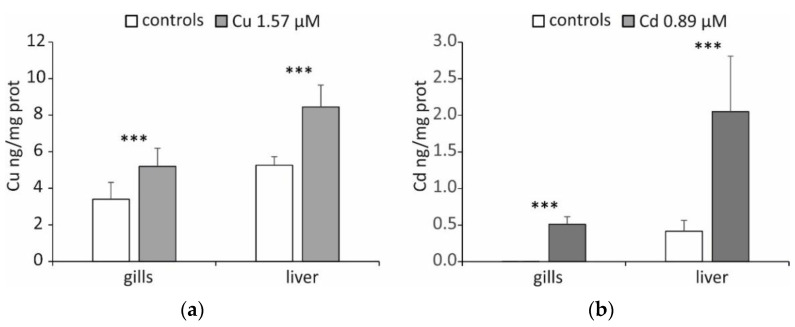
(**a**) Cu and (**b**) Cd concentrations (ng/mg of total proteins) determined in the gills and liver of *T. hansoni*. Data are reported as the mean of five specimens ± SD. Asterisks indicate differences between controls and treated specimens (*p* < 0.001).

**Figure 2 ijms-23-12799-f002:**
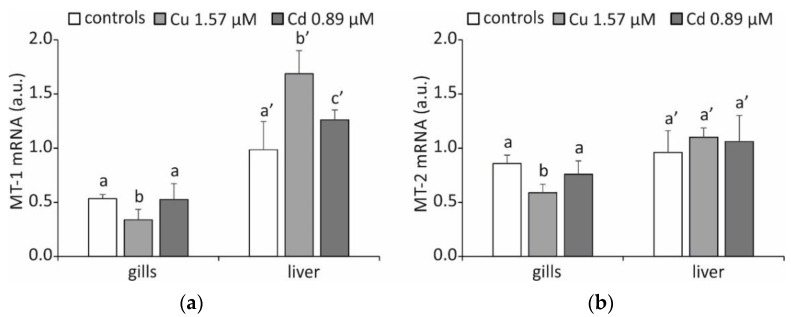
MT mRNA accumulation (arbitrary units) in gills and liver of *T. hansoni*. The results are expressed as the mean of five specimens ± SD. Different letters with the same index correspond to significant statistical differences among the different experimental conditions for *p* < 0.05. (**a**) MT-1; (**b**) MT-2.

**Figure 3 ijms-23-12799-f003:**
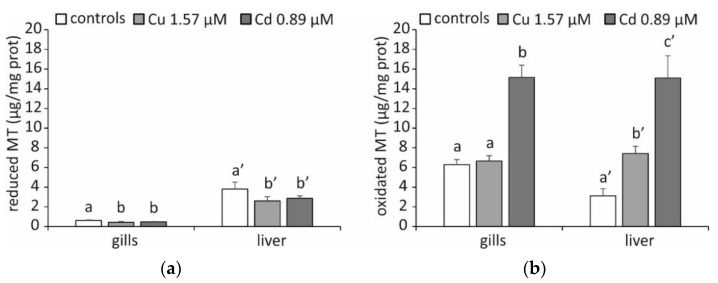
MT content (μg/mg total proteins) in gills and liver of *T. hansoni*. The results are expressed as the mean of five specimens ± SD. Different letters with the same index correspond to significant statistical differences among the different experimental conditions for *p* < 0.05. (**a**) Reduced MTs; (**b**) oxidized MTs.

## Data Availability

The data are contained within the article and Appendix A.

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
