# Peer review of "Metallothionein Expression as a Physiological Response against Metal Toxicity in the Striped Rockcod Trematomus hansoni"

_ijms, 2022, doi:10.3390/ijms232112799_

Round 1
Reviewer 1 Report (Previous Reviewer 3)
- A main point for this manuscript concerns section 4.7, Western blot Analysis (4. Materials and Methods). I have suggested adding the explanation for the method used to normalize the MT bands in the blot and on line 301-302, Authors added the sentence” On cell-free extracts, the protein concentration was determined as previously described in order to normalize the gel electrophoretic analysis”. This is not enough to normalize band signals in a blot and consequently to evaluate increase/decrease in protein quantity. The membrane should be also probed for a housekeeping protein to normalize bands of interest, MTs in this article, also considering that in chemiluminescence there may also be signal saturation problems. Please, explain this point. In addition, Figure 4 is a representative blot of how many blots? In other words, how many blot did you run? Please, add this information as well.
Other comments:
- At the beginning of the Results and Discussion section, it is now clear what Cu and Cd concentrations are used for the fish exposure experiments, but a brief explanation for the selection of these doses is still missing. Are these doses relevant to the environment? Have you tested the dose-response effects in a previous work? Please, add a brief explanation.
-Concerning figure 1b (my previous observation: it is not reported Cd-concentration in the gills, there isn’t the white bar in the figure), I suggest adding the explanation of the Authors in the main text of the manuscript “The Cd concentration in the gills of control specimens is really limited, and correspondently the height of the respective bar” or something like that.
-Figure 4: For figure 1b, it is not enough adding some molecular weight markers to implement figure 4 without further explaination. I have already suggested adding the full image as original image of the blot/membrane in the supplementary figure together with a molecular weight marker, that now is present in Figure 4. This is also an indication reported in the Instructions for Authors: “In order to ensure the integrity and scientific validity of blots (including, but not limited to, Western blots) and the reporting of gel data, original, uncropped and unadjusted images should be uploaded as Supporting Information files at the time of initial submission”. Please, clarify is different exposure times have been used for the marker and the samples to obtain a good quality of the chemiluminescent signals.
- For the commercial antibody used, mouse anti-metallothionein IgG1, the product code is not reported in the M&M section, please add this detail.
Author Response
- A main point for this manuscript concerns section 4.7, Western blot Analysis (4. Materials and Methods). I have suggested adding the explanation for the method used to normalize the MT bands in the blot and on line 301-302, Authors added the sentence” On cell-free extracts, the protein concentration was determined as previously described in order to normalize the gel electrophoretic analysis”. This is not enough to normalize band signals in a blot and consequently to evaluate increase/decrease in protein quantity. The membrane should be also probed for a housekeeping protein to normalize bands of interest, MTs in this article, also considering that in chemiluminescence there may also be signal saturation problems. Please, explain this point. In addition, Figure 4 is a representative blot of how many blots? In other words, how many blot did you run? Please, add this information as well.
Answer
We fully agree with the Reviewer that the Western blotting normalization method is not appropriate for a quantitative evaluation of MT expression. However, we carried out this assay only to have a qualitative confirmation of the quantitative results obtained with the biochemical MT assay. In fact, it was performed on the cell extract of only three specimens (one for each experimental condition) and on the liver only. Since the result obtained with this analysis is not actually discussed in the manuscript and does not add anything more than the biochemical analysis of the MT content, we have eliminated everything related to Western blotting analysis from the manuscript.
Other comments:
- At the beginning of the Results and Discussion section, it is now clear what Cu and Cd concentrations are used for the fish exposure experiments, but a brief explanation for the selection of these doses is still missing. Are these doses relevant to the environment? Have you tested the dose-response effects in a previous work? Please, add a brief explanation.
Answer
As indicated in the materials and methods (section 4.2), we used "sub-lethal doses previously used in similar experimentations on fish of the same genus". This choice was made to be able to make a comparison between different species exposed to the same concentrations of Cu and Cd. Furthermore, the goal was to produce a tissue accumulation of metal sufficient to produce a cellular response. Since the exposure time had to be necessarily reduced, due to the limited possibilities granted by the particular laboratory conditions, it was decided to use non-"environmental" concentrations. We have added this explanation on page, 7 lines 233-238.
-Concerning figure 1b (my previous observation: it is not reported Cd-concentration in the gills, there isn’t the white bar in the figure), I suggest adding the explanation of the Authors in the main text of the manuscript “The Cd concentration in the gills of control specimens is really limited, and correspondently the height of the respective bar” or something like that.
Answer
In reference to this, we have added on page 2, line 102 ", in which the Cd concentration is close to zero".
-Figure 4: For figure 1b, it is not enough adding some molecular weight markers to implement figure 4 without further explaination. I have already suggested adding the full image as original image of the blot/membrane in the supplementary figure together with a molecular weight marker, that now is present in Figure 4. This is also an indication reported in the Instructions for Authors: “In order to ensure the integrity and scientific validity of blots (including, but not limited to, Western blots) and the reporting of gel data, original, uncropped and unadjusted images should be uploaded as Supporting Information files at the time of initial submission”. Please, clarify is different exposure times have been used for the marker and the samples to obtain a good quality of the chemiluminescent signals.
Answer
For the reasons mentioned above, this part has been removed from the manuscript.
- For the commercial antibody used, mouse anti-metallothionein IgG1, the product code is not reported in the M&M section, please add this detail.
Answer
For the reasons mentioned above, this part has been removed from the manuscript.
Reviewer 2 Report (Previous Reviewer 2)
The research work of Bakiu et al. evaluates the possibility of Trematomus hansoni, an teleost, to face increased copper and/or cadmium concentrations.
I consider that the Abstract has a few inconsistencies, as follows:
Copper or cadmium? Should it be "and" instead of "or".
There is nothing to simulate the natural environment, furthermore the specimens were kept in laboratory conditions. Therefore either take out "in the environment" or reinforce the statement of laboratory conditions as in the conditions imitated natural conditions.
Introduction
Little is said about ROS production in fish and how Cu and Cd affect it.
Line 38: "constant temperature as well as a high oxygen concentration" citation(s) required.
Lines 46-47: "It is not an essential metal to biota, but biological processes also influence its distribution in the seawater column" another statement that needs citation(s).
Line 49: "Cd, unlike Cd" ?
Line 49-50: " is an essential metal for biota but can lead to toxicity if present at high concentrations." citation required.
Lines 58-60: "It is well known that metal ions widely interact with biological molecules. Cd has a high affinity for the sulfhydryl groups of cysteine and competes against zinc (Zn) and Cu for the structural and active sites of various enzymes, thus impairing their catalytic activities" Citation required
Lines 62-65: "On the other hand, essential trace metals, such as Cu and Zn, are required in various physiological and metabolic processes, but they become toxic at excessive concentrations, damaging plasma the membrane, and other cell components". Citations
"Cu is also a redoxactive metal and can act as a catalyst in the Fenton reaction, facilitating the conversion of superoxide anion and hydrogen peroxide to hydroxyl radical, the most dangerous among reactive oxygen species (ROS) and mainly responsible for oxidative stress [11]."
High dosages of copper enhanced GPx activities and MDA levels in most tissues (Winston, 1991) ([11]) . Indeed copper ions catalyzes the conversion of H(2)O(2) to hydroxyl radicals in vitro [1] but where exactly in your citation ( [11] ) have you seen being stated that Cu facilitates the conversion of superoxide anion and hydrogen peroxide to hydroxyl radical?
Perform a clean up regarding citation use, and add citations for important statements otherwise my next review report will be "reject".
Materials and methods "4.5. Estimation of metal and metallothionein concentrations" is unclear state the protocols in a step by step, clear manner.
1. Halliwell, B., & Gutteridge, J. M. (1990). Role of free radicals and catalytic metal ions in human disease: an overview. Methods in enzymology, 186, 1–85. https://doi.org/10.1016/0076-6879(90)86093-b
Author Response
The research work of Bakiu et al. evaluates the possibility of Trematomus hansoni, an teleost, to face increased copper and/or cadmium concentrations.
I consider that the Abstract has a few inconsistencies, as follows:
Copper or cadmium? Should it be "and" instead of "or".
Answer
Page1, line 18 “or” changed in “and”.
There is nothing to simulate the natural environment, furthermore the specimens were kept in laboratory conditions. Therefore either take out "in the environment" or reinforce the statement of laboratory conditions as in the conditions imitated natural conditions.
Answer
Page 1, lines 18-19 “an increase of copper and cadmium concentrations in the environment” changed in “an increase of copper and cadmium concentrations in its tissues”.
Introduction
Little is said about ROS production in fish and how Cu and Cd affect it.
Answer
The production of ROS in fish is not different from that occurring in other organisms, as well as the reactions that take place between Cd or Cu and biological molecules, which can favour the formation of ROS.
Page 2, lines 59-66. “Metal ions can cause severe problems in fish by producing reactive oxygen species (ROS), which can cause oxidative stress and damage DNA, proteins, and lipids [13]. Cu is one of the active redox components that promote the redox response to ROS. Cu can specifically operate as a catalyst in the Fenton reaction, promoting the transformation of superoxide anion and hydrogen peroxide into hydroxyl radical, the most damaging ROS and the primary cause of oxidative stress. [14]. On the other hand, re-dox inactive metals such as Cd show their toxic effects via bonding to sulphydryl groups of proteins and depletion of glutathione [15].”. Some citations have been also added.
Line 38: "constant temperature as well as a high oxygen concentration" citation(s) required.
Answer
A citation (Sidell, 2000) has been added.
Lines 46-47: "It is not an essential metal to biota, but biological processes also influence its distribution in the seawater column" another statement that needs citation(s).
Answer
The sentence has been deleted because this concept is also expressed later in the text.
Line 49: "Cd, unlike Cd" ?
Answer
The sentence has been deleted because this concept is also expressed later in the text.
Line 49-50: " is an essential metal for biota but can lead to toxicity if present at high concentrations." citation required.
Answer
The sentence has been deleted because this concept is also expressed later in the text.
Lines 62-65: "On the other hand, essential trace metals, such as Cu and Zn, are required in various physiological and metabolic processes, but they become toxic at excessive concentrations, damaging plasma the membrane, and other cell components". Citations
Answer
A citation (Grosell, 2012) has been added.
"Cu is also a redox active metal and can act as a catalyst in the Fenton reaction, facilitating the conversion of superoxide anion and hydrogen peroxide to hydroxyl radical, the most dangerous among reactive oxygen species (ROS) and mainly responsible for oxidative stress [11]."
High dosages of copper enhanced GPx activities and MDA levels in most tissues (Winston, 1991) ([11]) . Indeed copper ions catalyzes the conversion of H(2)O(2) to hydroxyl radicals in vitro [1] but where exactly in your citation ( [11] ) have you seen being stated that Cu facilitates the conversion of superoxide anion and hydrogen peroxide to hydroxyl radical?
Answer
We changed Winston, 1991 with Halliwell and Gutteridge, 1990.
Perform a clean up regarding citation use, and add citations for important statements otherwise my next review report will be "reject".
Answer
We have followed all the Reviewer’s suggestion related to citations. If the Reviewer have more suggestions we are available to follow them.
Materials and methods "4.5. Estimation of metal and metallothionein concentrations" is unclear state the protocols in a step by step, clear manner.
Answer
The Reviewer will certainly agree that the homogenization of a sample is not so difficult to understand, as it is a basic biochemical laboratory methodology. I can understand that readers that are unfamiliar with atomic absorption spectrophotometry may have difficulties reading the text, but we certainly cannot describe the physicochemical principles underlying it. Therefore, we have only briefly described the MT quantification method.
Page 7, lines 275-292. “MT concentration was determined in the cell-free extracts by the silver saturation method [33]. Briefly, 0.5 mL of supernatant was mixed with 1.0 mL of 0.5 M glycine buffer, pH 8.5, and 1.0 mL of 20 ppm AgNO3 in 0.5 M glycine buffer, pH 8.5. After standing at room temperature for 15 min, to lead Ag+ to saturate the metal binding sites of MT, 100 μl of bovine hemolyzed erythrocytes were added. Then, the mixture was boiled for 2 min to cause denaturation and precipitation of proteins (but non-thermostable MTs) and centrifuged at 4000 × g for 5 min to remove Ag+ bound to hae-moglobin and other proteins from the solution. The hemolysate addition, boiling, and centrifugation steps were repeated twice. The MT concentrations were calculated based on the amount of Ag+ that remains in the solution because bond to MTs. Ag+ con-tent was determined by atomic absorption spectroscopy as previously described. The working wavelength was 328.068. To discriminate between reduced and oxidized MTs, Santovito et al.'s method [34] was applied. Briefly, 3.0 mL of cell-free extract were mixed with 2-mercaptoethanol (a reducing agent) at a final concentration of 10 mM, incubated at room temperature for 15 min, and then 2h after adding ZnCl2 at a final concentration of 6 mM. Both these incubations were performed in anaerobic condi-tions, gurgling nitrogen through the solution. The MT concentrations were determined as previously described. As previously described, the amount of MTs was normalized against total soluble cell proteins.”.
Round 2
Reviewer 1 Report (Previous Reviewer 3)
the manuscript can be accepted in its current form
Author Response
Thank you very much.
Reviewer 2 Report (Previous Reviewer 2)
The authors have significantly improved their manuscript.
Minor issues have remained, as follows:
"Little is said about ROS production in fish and how Cu and Cd affect it.
Answer
The production of ROS in fish is not different from that occurring in other organisms, as well as the reactions that take place between Cd or Cu and biological molecules, which can favour the formation of ROS."
Incorrect, there are significant differences (1,2), please state them.
Little is said about fish but in mammals, ROS may not necessarily associate with chronic Cd exposure(3)
"Line 49: "Cd, unlike Cd" ?
Answer
The sentence has been deleted because this concept is also expressed later in the text."
This referred to the typo Cd unlike Cd, please don't use copy paste answers, where was the concept later expressed in text?
"Materials and methods "4.5. Estimation of metal and metallothionein concentrations" is unclear state the protocols in a step by step, clear manner.
The Reviewer will certainly agree that the homogenization of a sample is not so difficult to understand, as it is a basic biochemical laboratory methodology. I can understand that readers that are unfamiliar with atomic absorption spectrophotometry may have difficulties reading the text, but we certainly cannot describe the physicochemical principles underlying it. Therefore, we have only briefly described the MT quantification method.
Page 7, lines 275-292. “MT concentration was determined in the cell-free extracts by the silver saturation method [33]. Briefly, 0.5 mL of supernatant was mixed with 1.0 mL of 0.5 M glycine buffer, pH 8.5, and 1.0 mL of 20 ppm AgNO3 in 0.5 M glycine buffer, pH 8.5. After standing at room temperature for 15 min, to lead Ag+ to saturate the metal binding sites of MT, 100 μl of bovine hemolyzed erythrocytes were added. Then, the mixture was boiled for 2 min to cause denaturation and precipitation of proteins (but non-thermostable MTs) and centrifuged at 4000 × g for 5 min to remove Ag+ bound to hae-moglobin and other proteins from the solution. The hemolysate addition, boiling, and centrifugation steps were repeated twice. The MT concentrations were calculated based on the amount of Ag+ that remains in the solution because bond to MTs. Ag+ con-tent was determined by atomic absorption spectroscopy as previously described. The working wavelength was 328.068. To discriminate between reduced and oxidized MTs, Santovito et al.'s method [34] was applied. Briefly, 3.0 mL of cell-free extract were mixed with 2-mercaptoethanol (a reducing agent) at a final concentration of 10 mM, incubated at room temperature for 15 min, and then 2h after adding ZnCl2 at a final concentration of 6 mM. Both these incubations were performed in anaerobic condi-tions, gurgling nitrogen through the solution. The MT concentrations were determined as previously described. As previously described, the amount of MTs was normalized against total soluble cell proteins.”."
Although I admit I am not an expert in AAS I asked for the used methodology not for the principles of atoms and ions to absorb light. I am sorry, was there a problem with the request for the method to be described? The purpose of an research paper is to also be reproducible, as even small variation in methodology make this difficult. Therefore, it is common sense to either cite the method or state it clearly in a step-by-step manner instead of using "as previously described" when it was not actually described at all. Thank you for the clarifications added.
References
1. Wiens L, Banh S, Sotiri E, Jastroch M, Block BA, Brand MD and Treberg JR (2017) Comparison of Mitochondrial Reactive Oxygen Species Production of Ectothermic and Endothermic Fish Muscle. Front. Physiol. 8:704. doi: 10.3389/fphys.2017.00704
2. Wilhelm Filho D. (2007). Reactive oxygen species, antioxidants and fish mitochondria. Frontiers in bioscience : a journal and virtual library, 12, 1229–1237. https://doi.org/10.2741/2141
3. Liu, J., Qu, W., & Kadiiska, M. B. (2009). Role of oxidative stress in cadmium toxicity and carcinogenesis. Toxicology and applied pharmacology, 238(3), 209–214. https://doi.org/10.1016/j.taap.2009.01.029
Author Response
Incorrect, there are significant differences (1,2), please state them.
Little is said about fish but in mammals, ROS may not necessarily associate with chronic Cd exposure(3)
Answer
Maybe there was a misunderstanding. Certainly, the amount of ROS produced and the production kinetics are different between one species and another (but also between different organs of the same species). My sentence referred to the modality of formation, that is, to the chemical mechanisms. And also in the review by Danilo Wilhelm Filho it is written: “The mechanism of fish mitochondrial superoxide anion (O2•-) and ROS production as well as the mechanism of mitochondrial coupling and proton leak seems similar to that of mammals.”.
It is absolutely true that the formation of ROS is not exclusively a consequence of the presence of exposure to Cd. We have added the following text. Lines 60-65. “These substances are a normal byproduct of oxidative metabolism, and generally, a higher rate of ROS generation has been reported in aquatic invertebrates and in fish [13]. On the other hand, mitochondria of true endotherms, such as mammals, seem to produce lower amounts of ROS than fish [14]. However, the rate of ROS formation can vary from species to species, and also greatly increase in response to various physiological and pathological conditions,”. We also cited the papers that the Reviewer suggested.
This referred to the typo Cd unlike Cd, please don't use copy paste answers, where was the concept later expressed in text?
Answer
Yes, I understood that, but I just wanted to specify that the correction was not necessary because later in the text there is the following sentence. Lines 55-58 “Cd also exerts several toxic effects at both cellular and systemic levels [11]. On the other hand, essential trace metals, such as Cu, are required in various physiological and metabolic processes, but they become toxic at excessive concentrations, damaging plasma membrane, and other cell components [12]. ".
Although I admit I am not an expert in AAS I asked for the used methodology not for the principles of atoms and ions to absorb light. I am sorry, was there a problem with the request for the method to be described? The purpose of an research paper is to also be reproducible, as even small variation in methodology make this difficult. Therefore, it is common sense to either cite the method or state it clearly in a step-by-step manner instead of using "as previously described" when it was not actually described at all. Thank you for the clarifications added
Answer
No, there is absolutely no problem with the request, and I apologize if my answer seemed controversial. The fact is that the AAS analysis of a cell-free extract does not require any preliminary operation (unlike what happens for a tissue or for cells in culture) other than the setting and calibration of the instrument with the parameters that are indicated in the text. Once this is done, the instrument draws the sample into the atomization chamber and the metal concentration is read. It seemed inappropriate to us to write this in the text, but if the Reviewer finds it useful, I will add it without problems.
This manuscript is a resubmission of an earlier submission. The following is a list of the peer review reports and author responses from that submission.
Round 1
Reviewer 1 Report
1. Authors should consider to include more details in the experimental section, specially for the results of the Western blot as seen in Figure 4 and in sections 4.5 and 4.7 of their manuscript.
Reviewer 2 Report
In this work, Bakiu et al. investigated metal bioaccumulation and metallothionein (MT) expression in Trematomus hansoni gills and liver under the light of future metal exposure the species might face in the future in the context of increased water pollution.
Introduction
There is little to no data regarding Trematomus hansoni physiology and oxidative stress response. This data needs to be thoroughly added.
4.7. Western blot Analysis
Molecular weight markers are lacking (Fig 4) also use an software (such as Image J) to edit the image into an acceptable form.
4.5. Estimation of metal and metallothionein concentrations
Add chromatograms if available.
The atomic absorption spectroscopy is unclear, state the protocol in a step by step clear manner.
4.2. Experimental animals
State the exposure conditions
State the specimen weight and size, the evaluation of adulthood was made accordingly to what criteria?
What was the size of the water tanks (in Liters)
Also state the water pH and oxigen levels (basic parameters that influence metabolism therefore the heavy metal absorbtion).
Conclusions
Line 278: "the escribed toxicological evaluations a functional instrument for the previous and initial monitoring of the ecosystem of the new MPA" - There is a typo "e" is missing. The work has insufficient data to aim at being an instrument for monitoring the ecosystem, please rephrase.
"Funding: This research received no external funding. This research was supported by the Italian 293 National Program for Antarctic Research (PNRA)." It is unclear, the research received funding from the national program or not?
Reviewer 3 Report
The paper reports on metallothionein gene expression as well as its protein transduction on the fish Trematomus hansoni (an Antarctic teleost widespread in the coastal marine areas of the ice continent) following laboratory exposure to copper or cadmium taking into consideration that in Antarctic seawater there are a natural occurrence of high cadmium (Cd) and copper (Cu) concentrations. The role played by MT is analyzed using different technical approach.
Specific comments:
-At the beginning of the Results and Discussion section it is not clear the concentration of Cu and Cd used for fish exposure experiments. They are reported later and are described in Material and Methods section, but I suggest introducing this information in the first part of the Results and Discussion, section together with a brief explanation for the selection of these doses.
-Figure 1: In figure 1b, it is not reported Cd-concentration in the gills, there isn’t the white bar in the figure, please check.
-Line 206: it is reported a concentration of “Cu (1.57 M)”, is it correct? please check.
-Figure 4: In figure 1b, it is reported the MT band by Western blotting, I suggest adding the full image as original image of the blot/membrane in the supplementary figure together with a molecular weight marker, absent in Figure 4.
Concerning the commercial antibody used, mouse anti-metallothionein IgG1, I suggest adding some information to its cross reactivity. Do it recognize MT1 and MT2 isoforms? In addition, the molecular weight of the MT bands is not reported, please add this information. Small differences in MT molecular weights may be possible in different species.
In the section 4.7. Western blot Analysis (4. Materials and Methods), I suggest adding the explanation for the method used to normalize the MT bands in the blot